# Impact of Periodontal Attachment Loss on the Outcome of Endodontic Microsurgery: A Systematic Review and Meta-Analysis

**DOI:** 10.3390/medicina57090922

**Published:** 2021-09-01

**Authors:** Margarida Sarnadas, Joana A. Marques, Isabel Poiares Baptista, João Miguel Santos

**Affiliations:** 1Institute of Endodontics, Faculty of Medicine, University of Coimbra, 3000-075 Coimbra, Portugal; uc2015238581@student.uc.pt (M.S.); joanaamarques@uc.pt (J.A.M.); 2Institute of Periodontology, Faculty of Medicine, University of Coimbra, 3000-075 Coimbra, Portugal; icbaptista@fmed.uc.pt

**Keywords:** apicoectomy, endodontic microsurgery, endodontic-periodontal lesion, isolated endodontic lesion, outcome, prognostic factors

## Abstract

*Background and Objectives*: Endodontic microsurgery (EMS) aims to eradicate the sources of infection once the apical root resection removes most of the infected anatomical structures and repairs potential procedural errors in the apical region. An endodontic-periodontal lesion yields a pathological communication between the pulp and the periodontium. The purpose of this systematic review and meta-analysis is to evaluate the impact of periodontal attachment loss on the outcome of teeth submitted to EMS. *Materials and Methods*: PRISMA guidelines were followed. An electronic search was performed in EBSCOhost, Embase and PubMed databases with the following search key: (“endodontic microsurgery” AND “outcome”). No filters were used concerning the year of publication or language. Only randomized clinical trials, prospective and retrospective clinical studies in humans, with a minimum one-year follow-up, defined clinical and radiographic outcome criteria and estimable success rate for endodontic-periodontal lesion were included. Statistical analysis was performed using OpenMeta[Analyst] software. *Results*: Of a total of 113 articles, 34 were selected for full-text reading after duplicates deletion and title and abstract analysis. Thirteen and six studies were included in the systematic review and meta-analysis, respectively. A total of 2775 pooled teeth were submitted to EMS, of which 492 teeth and 4 roots had periodontal involvement. According to the qualitative analysis, success rates of the endodontic-periodontal group ranged from 67.6% to 88.2%. Meta-analysis revealed that the absence of periodontal attachment loss was predictive of a higher likelihood of success with an odds ratio of 3.14. *Conclusions*: Periodontal attachment loss presents a risk factor for EMS outcome. Although endodontic-periodontal lesions were associated with lower success rates considering a 1 to 10 years follow-up period, long-term successful prognosis following EMS has been reported, therefore presenting a fully valid and viable therapeutic option for the management of this type of lesions.

## 1. Introduction

The main goal of endodontic treatment is to prevent or cure apical periodontitis (AP). The exposure of vital pulp to different microorganisms results in infection of the root canal system, consequently inducing pulp tissue necrosis and infection growth in the periapical region. Host’s immune response is then activated, with local inflammation taking place, as well as periapical tissues’ resorption and destruction, ultimately leading to periapical lesions formation and AP establishment [1,2]. An increase in worldwide prevalence of AP in the general adult population has been reported, especially among people over 50 years old [2]. Previous studies indicate that 33% to 60% of root-filled teeth present AP, due to the primary infection or emergence of a secondary infection [1,3]. The main causes for root canal treatment failure include extraradicular infection, foreign body reaction, and root canal system persistent infection caused by complex anatomical structures at the periradicular area and periradicular cysts. However, several iatrogenic factors may also be related to post-treatment endodontic disease, such as root perforation, ledge formation, instrument fracture and overfilling [4,5].

Nonsurgical endodontic retreatment (NSER) remains the desirable treatment option to manage post-treatment AP, avoiding tooth extraction and dental implant placement, and allowing natural tooth and alveolar ridge preservation [1,6]. Nevertheless, surgical endodontic retreatment (SER) is also indicated to eradicate post-treatment AP, as a last resort treatment when NSER is considered unfeasible, failed, or unlikely to improve the previous condition [5,7,8,9,10,11,12].

Endodontic microsurgery (EMS) aims to eradicate the sources of infection once the apical root resection removes most of the infected anatomical structures and repairs potential procedural errors in the apical region [4,11]. This surgical procedure is characterized by the use of an operating microscope which improves illumination and magnification, thus allowing to meticulously identify apical anatomy and examine the resected root surface [6,7,13]. EMS also demands the use of ultrasonic devices to perform root-end preparation. Additionally, root-end filling encompasses the placement of biocompatible materials which ensure a hermetic seal of the root canal system and enable healing as it forms an apical barrier between the affected root and surrounding tissues [7,11,13,14,15,16]. These surgical advancements allow EMS to be executed with precision and predictability, eliminating the difficulties associated with traditional endodontic surgery such as poor visualization, inaccurate root-end preparation and large osteotomy [15,16,17]. EMS success rate increased from 59% to 94% after the introduction of modern surgical techniques [18].

Periodontitis is defined as a chronic multifactorial inflammatory disease triggered by dysbiotic subgenvival biofilm that gradually promote the destruction of the tooth’s supporting structures, including alveolar bone and periodontal ligament. Periodontal attachment loss is diagnosed by clinical attachment loss, periodontal pocket depth, bleeding on probing, and radiographic alveolar bone loss [19]. Severe periodontitis is the sixth most prevalent disease worldwide, with a prevalence of 11.2% and over 743 million affected people, significantly impairing quality of life as it may lead to tooth loss and considerable masticatory function compromise [19,20,21]. Moreover, systemic health repercussions may occur. Furthermore, the current population aging is expected to become associated with an increase in the prevalence of periodontal attachment loss [20,21].

An endodontic-periodontal lesion yields a pathological communication between both pulp and periodontal tissue through the apex, lateral canals, and/or dentinal tubules [19,22,23]. Regarding EMS in such clinical diagnosis, two scenarios may occur: the tooth subjected to the procedure may be posteriorly affected by periodontal attachment loss, or a tooth exhibiting periodontal disease can undergo EMS [24]. In either scenario, EMS decreases root length, thus altering the crown-to-root ratio (CRR) and periodontal support. Also, this procedure modifies the tooth’s biomechanical response, causing unfavorable stress distribution and increased tooth mobility, which may influence tooth function and survival as it remains exposed to continuous occlusal loading [4,25]. Periodontal bone loss also aggravates CRR, simultaneously increasing the clinical crown length and decreasing the supported root area. Since the functional stress is mostly concentrated on the cervical root third, periodontal bone loss has a greater influence on biomechanical parameters than the apical root resection itself [4,24,25]. Moreover, as mentioned above, patients’ occlusion also impairs tooth stability after EMS. [4].

Kim and Kratchman proposed a diagnosis classification of lesion types previous to EMS into A-F categories [12]. This classification includes periodontally involved teeth (categories D, E and F), therefore being the most widely accepted.

Most studies on the prognosis of EMS show high success and survival rates, thus attesting this procedure’s effectiveness. However, few studies include teeth with periodontal involvement, which is suggested to adversely impact the outcome of EMS [7,11,13,15,26,27]. Considering the high prevalence of teeth showing periodontal attachment loss submitted to EMS, as well as the lack of scientific evidence on the topic, it is crucial to clarify the impact of periodontal involvement on the outcome of this surgical approach. To our knowledge, the present study consists of the first systematic review and meta-analysis addressing this issue. 

Thus, the purpose of this systematic review and meta-analysis is to evaluate the impact of periodontal attachment loss on the outcome of teeth submitted to EMS.

## 2. Materials and Methods

PRISMA (Preferred Reporting Items for Systematic Reviews and Meta-Analysis) guidelines were followed. The research question was defined according to the paradigm of evidence-based dentistry, following the PEO (Population, Exposure and Outcome) format for systematic reviews of risk suggested by the Joanna Briggs Institute: Population: teeth with periodontal attachment loss.Exposure: EMS.Outcome: clinical and radiographic success.

This review aimed to answer the following questions: “Do teeth with periodontal attachment loss submitted to EMS present poorer clinical and radiographic outcome? Are those associated with higher risk of failure?”

### 2.1. Searching Criteria

The selection of studies for this systematic review was based on the inclusion and exclusion criteria shown in Table 1.

The included studies pursued Rud et al. [28] and Molven et al. [29] criteria for assessing of healing outcomes, with categories as follows: complete healing (re-establishment of the lamina dura); incomplete healing (the former radiolucency decreased in size or remained stationary with an irregular periphery); uncertain healing (the former radiolucency decreased in size or remained stationary with a circular periphery); and unsatisfactory healing (the former radiolucency increased in size). Outcome was dichotomized into success, when complete or incomplete healing was attributed, and failure, when healing was uncertain or unsatisfactory.

### 2.2. Searching Method

An initial search limited to the Journal of Endodontics was conducted to gather topic-related studies, regardless of the publication type. The collected information helped to develop a search strategy, particularly concerning the identification of keywords and index terms. Afterwards, three electronic databases were used: EBSCOhost, Embase, and PubMed. The following search key was used in each database: (“endodontic microsurgery” AND “outcome”). The electronic search was then complemented with a manual search, by checking the references of the most relevant papers.

### 2.3. Study Selection

The study selection ended in April 2021, with no restrictions in regard to language. The obtained articles were individually scanned by two reviewers (M.S. and J.M.S.). After duplicates deletion, two researchers (M.S. and J.M.S.) independently screened the title and abstract of each article in order to assess its eligibility, excluding those which did not meet the main subject. Subsequently, the included studies were subjected to a full-text evaluation to identify those that meet the previously defined inclusion criteria. In case of disagreement during the selection process, a third examiner (I.P.B.) was consulted.

### 2.4. Data Extraction

Two authors (M.S. and J.M.S.) independently participated in the data extraction process. General information from each of the selected articles was collected to create a table of evidence. An Excel^®^ (Microsoft Corporation, Albuquerque, NM, USA) table containing the following data was created: study identification (title, authors, and digital object identifier), year of publication, study design, sample size, tooth type, diagnostic criteria of periodontal attachment loss, measure unit, anesthesia type, additional hemostasis strategies, root filling material, regeneration materials, follow-up period, recall rate, success rates for both lesion type groups (isolated endodontic and endodontic-periodontal lesions).

### 2.5. Quality Assessment

The methodological quality evaluation of the eligible studies was conducted prior to inclusion in this review. Quality assessment was performed by two independent authors (M.S. and J.M.S.). Two Cochrane risk-of-bias assessment tools were applied: RoB 2, for the randomized controlled trial, and ROBINS-I, for both prospective and retrospective cohort studies). Newcastle-Ottawa Scale was also used to assess prospective and retrospective cohort studies.

### 2.6. Meta-Analysis

The relevant data of the studies included in the qualitative analysis was extracted. Descriptive analysis was used to identify similarities and variations between the studies. Only studies that followed the Kim and Kratchman classification [12] as diagnostic criteria of periodontal attachment loss were considered for meta-analysis. This classification is entirely described in Appendix A.

Studies were pooled in a statistical meta-analysis of proportions with difference of arcsines transformation using OpenMeta[Analyst] software. Heterogeneity was statistically assessed using the standard Chi-square and I-square tests. Statistical analysis was then performed using DerSimonian-Laird binary random-effects at a confidence interval of 95%.

## 3. Results

### 3.1. Study Selection

The flowchart according to PRISMA guidelines is provided in Figure 1.

A total of 219 articles were obtained in the electronic search, and 2 additional articles were identified through manual search. After duplicates removal, 113 articles remained. The titles and abstracts of the 113 selected articles were screened and 79 articles were excluded. Then, the remaining 34 articles were subjected to full-text analysis to assess eligibility, with 21 articles being excluded [30,31,32,33,34,35,36,37,38,39,40,41,42,43,44,45,46,47,48,49,50] according to the reasons listed in Appendix A. Therefore, 13 articles were included in the qualitative assessment: one randomized controlled trial [8], four prospective cohort studies [13,14,15,51], and eight retrospective cohort studies [9,10,11,17,26,52,53,54]. For the quantitative analysis, only six articles were included: two prospective cohort studies [13,15] and four retrospective cohort studies [9,11,26,54].

### 3.2. Study Characteristics

Table 2 presents the detailed data concerning the studies included in this review. 

Of a total of 2775 pooled teeth submitted to EMS, 492 teeth and 4 roots had periodontal involvement. All studies included anterior teeth, premolars and molars, although none specified the distribution of the tooth type of the endodontic-periodontal group. 

The diagnostic criteria of endodontic-periodontal lesions differed between studies: some resorted to the classification proposed by Kim and Kratchman [9,11,13,15,26,54], others used the criteria of marginal bone loss or probing depth above 3 mm [10,14,51,52,53], alveolar dehiscence [8], and periodontal involvement [17].

The tooth was always considered as the unit of evaluation, except for one study, in which the root was assessed as a single unit [52]. 

The anesthetic protocol mostly included 2% lidocaine and 1:80,000 epinephrine. However, two studies used 2% lidocaine and 1:100,000 epinephrine [8,17] and other two 4% articaine and 1:100,000 epinephrine [14,53]. For additional hemostasis, epinephrine, ferric sulphate or aluminum chloride were used in some cases [9,10,13,14,15,17,26,51,52,53,54].

Regarding the retrofilling material, the majority of the studies applied zinc oxide-eugenol intermediate restorative material (IRM) [9,10,11,13,15,26,52], mineral trioxide aggregate (MTA) [8,9,10,11,13,14,15,17,26,51,52,53,54] and super ethoxy-benzoic acid (SuperEBA) [9,11,13,14,15,26,51,53].

Six studies applied collagen resorbable membranes and/or bone substitutes for guided bone regeneration procedures [8,9,10,15,17,52].

The follow-up period ranged from 1 year [8,11,14,26] to a maximum of 10 years [9,13]. The maximum recall rate was 100% in two studies [11,54] and the minimum was 34.5% [17].

Concerning the endodontic-periodontal group, the higher success rate was 95.7% [9] and the lowest was 33.3% [17]. However, these outcomes validity might be compromised by a small sample size [17,52] and by the fact that Song et al. [9] has been considered an outlier. Therefore, the success rates for the endodontic-periodontal group ranged from 67.6% to 88.2% [8,10,11,13,14,15,26,51,53,54]. Most studies had a success rate of the endodontic group higher than the endodontic–periodontal group, disregarding two studies which demonstrated that teeth periodontally involved had greater success rates than those not affected [9,14].

Six studies [8,9,15,17,52,53] report results with a breakdown of the four categories of outcome (complete, incomplete, uncertain and unsatisfactory healing), but in seven studies the success is reported without differentiation of complete and incomplete healing [10,11,13,14,51,54]. Incomplete healing reported in reviewed studies ranged from 5% [9,52,53] to 15% [17] and over 20% [8].

### 3.3. Quality Assessment

Only one randomized controlled trial was included in this systematic review [8], which final risk of bias was defined as “low” according to the Cochrane RoB 2 tool (Appendix A). The 12 remaining prospective and retrospective cohort studies were assessed by the Newcastle–Ottawa Scale (Appendix A), with two studies [17,52] scoring six stars, one study [10] scoring seven stars and the other nine studies [9,11,13,14,15,26,51,53,54] scoring eight stars out of a maximum of nine stars. The same twelve studies were assessed by the Cochrane ROBINS-I tool (Appendix A), with the risk of bias being assessed as “low” for seven [9,11,13,14,15,52,53,54], “moderate” for two [26,51], and “high” for three [10,17,52] studies.

### 3.4. Meta-Analysis

The meta-analysis showed a clear tendency to favor the endodontic group, as only one study [9] presented higher success rates for the endodontic-periodontal group. The absence of periodontal attachment loss was predictive of a higher likelihood of success with an Odds Ratio of 3.14 (95% confidence interval: 2.023 to 4.870), as shown in Figure 2.

## 4. Discussion

EMS aims to eliminate the entire necrotic tissue from the surgical site and provide an adequate apical sealing, consequently allowing hard and soft tissues’ integrity restauration and reestablishment of the dentogingival complex [15].

Regarding the follow-up period, surgical retreatment cases are prone to heal faster than nonsurgical ones [55]. Song et al. [9] demonstrated that the most relevant evidence concerning the healing process was obtained at the first-year post-surgery and that the variation in the clinical outcome between one and four or more years follow-up period was not significant. Hence, the one-year follow-up may be sufficient to predict long-term outcome of EMS [8]. Therefore, the present systematic review established a minimum follow-up period of one year for study inclusion, resulting in studies ranging from 1 to 10 years follow-up period of evaluation.

The effect of the root-end filling material is one of the intraoperative key factors of EMS outcome. EMS requires biocompatible materials such as IRM, Retroplast, SuperEBA, MTA, among others [26]. MTA is the preferred EMS root-end filling material in most of the studies included in this systematic review [10,14,15,26,52,53]. MTA has the ability to stimulate bone, dentin, and cementum formation, promoting tissue regeneration (e.g., periodontal ligament and cementum) [15]. Von Arx et al. [53] also suggested that the most effective seal over a follow-up period of five years was achieved with MTA. However, Zhou et al. [8] found no significant difference in EMS clinical outcomes when comparing MTA and BP-RRM, with both showing favorable biocompatibility, no cytotoxic effects and similar sealing performance. However, one study [13] found no significant influence in the success rate regarding the root-end filling material. The remaining three studies [11,17,51] did not evaluate the effect of the root-end filling material on the outcome of EMS.

Periodontitis is responsible for alveolar bone and periodontal ligament loss, as well as apical migration of epithelial root adhesion, which may jeopardize the healing process after EMS. Therefore, the prognosis of periodontally involved teeth relies on both periodontal support and surgical approach [9,17,26]. Endodontic-periodontal lesions are thus one of the most challenging scenarios in SER field [24]. A tooth may have independent or communicating endodontic and periodontal lesions. Combined lesions may initially present as isolated endodontic or periodontal lesions, with subsequent involvement of one another [24].

EMS is considered a high success procedure, although it usually covers endodontic lesions without any periodontal complications [15], once endodontic-periodontal lesions are believed to have a worse prognosis when compared to isolated endodontic lesions [11,56]. However, as previously mentioned, in the regular clinical practice settings, many cases show some degree of periodontal involvement [15]. Therefore, the lesion type seems to be a significant outcome predictor [13,26,54]. Lui et al. [10] concluded EMS prognosis may not be influenced by the presence of buccal alveolar bone dehiscence. In accordance, von Arx et al. [57] analyzed the effect of bone defects size on EMS healing outcome, reporting that marginal bone loss was not significantly associated with healing at one year reassessment. Conversely, Song et al. [51] findings identify buccal bone plate height as the only factor among periapical defects that actually influenced the healing outcome, therefore concluding that the marginal bone deficiency resulted in a greater impairment of EMS outcome than periapical bone deficiency [51].

Concerning tooth type, the impact of periodontal attachment loss in EMS prognosis is believed to differ between single and multi-rooted teeth. As aforementioned, periodontal bone loss aggravates CRR, with the decrease of the supported root length being accompanied by the increase of the clinical crown length [4,24,25]. In a single-rooted tooth, periodontal bone loss has a greater influence on biomechanical parameters than the apical root resection itself. Stress resulting from occlusal loading is mostly concentrated at the cervical third of the root rather than at the apical region [4,24,25]. In a multi-rooted tooth, the bone loss at the apical level will not affect the prognosis as unfavorably as if it occurred at the cervical level, once the volume at the cervical level that the tooth occupies is more significant than at the apical portion. However, none of the studies included in this systematic review specified the distribution of tooth type within the group with periodontal involvement. For this reason, it is not possible to draw conclusions about the possible influence of single versus multi-rooted teeth yielding periodontal attachment loss on EMS prognosis.

The patient’s occlusion also has a great impact on tooth stability after EMS. In all occlusal relationships, the stress and tooth displacement maximum values at the cervix, root apex, alveolar bone, and periodontal ligament increased as the resection length increased [4]. Thus, EMS prognosis may differ among different occlusal relationships. Ran et al. showed greater stress and tooth displacement maximum values with increased overjet, followed by normal occlusions and increased overjet with deep overbites. Deep overbites had the lowest values [4].

A poor prognosis may result from the formation of a long junctional epithelium over the dehisced root surface since alveolar bone loss promotes the apical migration of gingival epithelial cells. The long junctional epithelium serves as a pathway for microorganisms dissemination, preventing the healing process which may lead to EMS failure [7,13,17,51]. To mitigate such negative outcome, some studies perform regeneration techniques such as guided tissue regeneration, aiming to potentiate EMS prognosis in endodontic-periodontal lesions [7,8,11,15]. Six studies [8,9,10,15,17,52] resorted regeneration techniques in this type of lesions. The most frequently applied materials were collagen resorbable membranes (e.g., CollaTape^®^ and BioMend^®^) and/or bone substitutes (e.g., BioOss^®^). Kim et al. [15] associated calcium sulfate to CollaTape^®^. The former material is extremely biocompatible, simple and effective [15]. Several studies hypothesized that combining guided tissue regeneration with EMS may not be mandatory in teeth with intact alveolar bone [10]. However, it is expected to improve the healing outcome in teeth presenting “through and through” lesions [10] or complete buccal bone dehiscence (class F lesions according to Kim and Kratchman’s classification), as confirmed by Zhou et al. [8] and Song et al. [9].

Assessment of EMS success relies on both radiographic resolution of the periapical radiolucency and absence of clinical symptoms [37]. The studies included in this systematic review follow the criteria established by Rud et al. [28] and Molven et al. [29] for healing classification. In the present review, outcome was dichotomized into success, when complete and incomplete healing was attributed, and failure, when healing was uncertain or unsatisfactory. In Kim et al. study [15], endodontic-periodontal lesions (classes D, E, and F according to Kim and Kratchman’s classification) showed success rates of 77.5%, whereas classes A, B, and C evinced a 95.2% success rate, in two to five years follow-up. The high success rate regardless of the lesion type may be related to EMS advantages and/or the use of regeneration techniques [15]. However, the lower success rate of endodontic-periodontal than isolated endodontic lesions could lead to the assumption that endodontic-periodontal lesions show more failed cases over time. Notwithstanding, Song et al. [9] verified that, among the seven failure cases with long-term follow-up, only one had periodontal involvement. This study was one of the two included studies which presented a higher success rate for endodontic-periodontal lesions over the isolated endodontic ones. In a different study, Song et al. [51] excluded a subgroup of 27 teeth with complete loss of the buccal bone plate (with marginal bone loss >3 mm) while evaluating the impact of marginal bone loss on the outcome. As a result, the success rate of teeth with marginal bone loss greater than 3 mm was overestimated. If the referred subgroup, with eight reported failures, was added to the 33 cases of marginal bone loss greater than 3 mm, the success rate would decrease from 87.9% to 80%. As for the studies of Huang et al. [52] and Yoo et al. [17], the low success rates of 50% and 33.3%, respectively, can be explained by the reduced sample size (four roots and nine teeth, respectively). The validity of the reported lower success rate is weakened by a low recall rate, as well as a considerable risk of bias.

Concerning the study of von Arx et al. [14], in which a higher success rate was reported for the periodontally involved teeth, it is worth mentioning that this is a preceding study to the one published in 2021 by the same group [53], which reported a lower success rate for the endodontic-periodontal group. We hypothesize that opposing conclusions derive from the difference in the follow-up period (1 vs. 5 year). Also, we believe a longer follow-up time [53] to provide more relevant data than the one-year control [14]. This evidence highlights the importance of a sufficiently long follow-up period to detect the outcome of interest, and echoes that healing peaks in the first year after EMS, and a reversal to disease occurs in 5% to 25% of the apparently healed cases within four years after treatment [58].

In regard to the limitations of this systematic review, the first aspect to point out is the lack of geographical variability of the studies. The included 13 studies correspond to only 6 different research teams: one from Switzerland [14,53] and the remaining from Asian countries (Singapore [10,52], Korea [9,11,13,15,17,26,51,54] and China [8]). Therefore, the obtained results should be carefully evaluated as they may not reflect the worldwide effectiveness of the intervention under study. Secondly, the majority of studies have brought together cases from the Department of Conservative Dentistry of Yonsei University, in Seoul (Korea) [9,11,13,15,26,51,54]. Kim et al. study [15] presents as the starting point of all studies performed by this research team, presenting strong evidence to assume that the database may be the same for all seven studies. Thus, it is very likely that there will be a sample overlap of these studies once some are follow-ups studies. Furthermore, as mentioned above, it is possible to verify this last aspect in von Arx’s studies, since the 2012 study [53] corresponds to a five-year follow up of the initial study [14]. Additionally, all studies included in the present review provide scientific evidence rendered in academical clinical settings. Aiming at reflecting the effectiveness of interventions in real-life routine conditions in general population, pragmatical clinical trials may be beneficial, with inclusion of patient reported outcomes to improve patient-clinician communication and the therapeutic decision process. Lastly, future clinical studies should be conducted to evaluate the influence of tooth type, single- or multi-rooted teeth, as well as occlusal relationships, in teeth with periodontal attachment loss on EMS prognosis.

## 5. Conclusions

This systematic review confirmed that isolated endodontic lesions were associated to higher success rates comprised between 78.2% and 95.3%, whereas endodontic-periodontal lesions ranged from 67.6% to 88.2%, considering a 1 to 10 years follow-up period.

Based on studies strictly following Kim and Kratchman’s diagnostic criteria, meta-analysis revealed periodontal attachment loss to be a risk factor for EMS outcome, with an odds ratio of 3.14. Although endodontic-periodontal lesions were associated with lower success rates, long-term successful prognosis following EMS has been reported, therefore presenting a fully valid and viable therapeutic option for the management of this type of lesions.

## Figures and Tables

**Figure 1 medicina-57-00922-f001:**
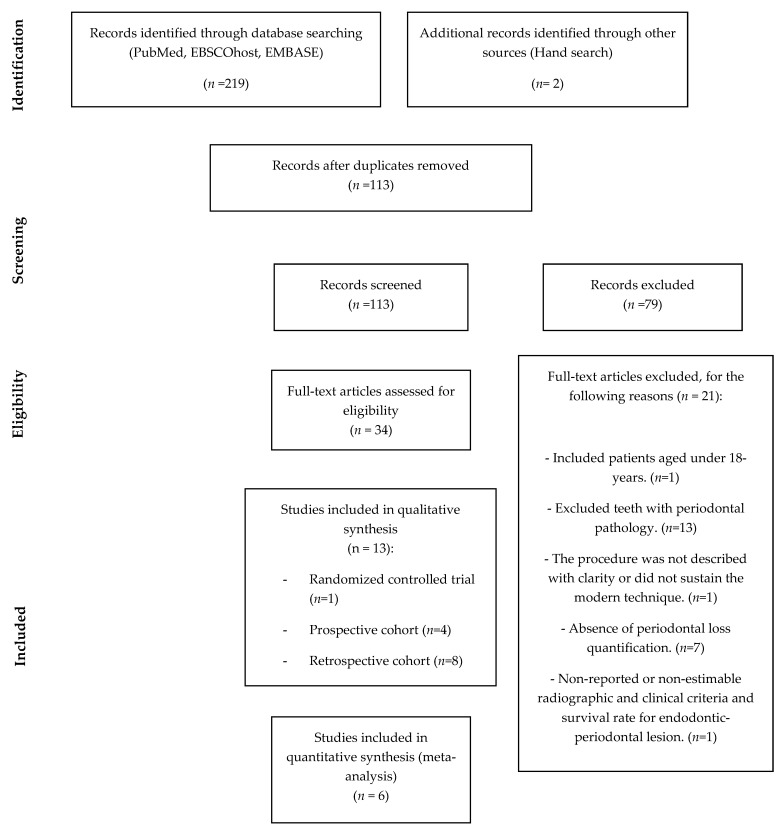
Flowchart of article selection method according to PRISMA Statement.

**Figure 2 medicina-57-00922-f002:**
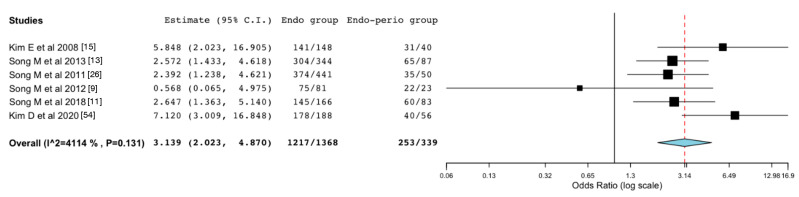
Forest plot of odds ratio of success in endodontic (endo) group compared to the endodontic-periodontal (endo-perio) group. Confidence interval (CI).

**Table 1 medicina-57-00922-t001:** Inclusion and exclusion criteria.

Inclusion criteria	1. Clinical studies in humans
	2. Randomized clinical trials (RCT)
	3. Prospective clinical studies
	4. Retrospective clinical studies
	5. Teeth with indication for EMS (periapical lesion, post-treatment apical periodontitis, extrusion of root canal filling material resulting from primary endodontic treatment, persistent extra-radicular infection)
	6. Studies in which the surgical procedure was detailed or sustained the modern technique by using magnification devices (microscope and endoscope) and ultrasonic root-end preparation
	7. Clinical and radiographic success following Rud and Molven’s criteria [28,29]
	8. Reported or estimable clinical and radiographic success rate for both isolated endodontic and endodontic-periodontal groups
	9. Minimum follow-up period of one year
	10. Quantified periodontal attachment loss
Exclusion criteria	1. Patients aged under 18 years
	2. Exclusion of teeth with periodontal attachment loss
	3. Systematic review
	4. Case series
	5. Case report
	6. The surgical procedure was not detailed or did not sustain the modern technique
	7. Unclear clinical and radiographic success criteria
	8. Non-reported or non-estimable endodontic-periodontal lesion success rate
	9. < One-year follow-up
	10. Absence of periodontal attachment loss quantification

**Table 2 medicina-57-00922-t002:** Summary of the studies included in the systematic review.

Study	Study Design	Sample Size (Teeth)			Diagnostic Criteria of Periodontal Attachment Loss	Root Filling Material	Follow-Up Period (Years)	Recall Rate (%)	Success Rate (%)		Regeneration	
		*n* Initial	*n* Final	*n* EP					Endo-Group	EP-Group	Yes/No	Material
Zhou et al., 2017 [8]	Randomized Controlled Trial	240	158	17	Alveolar dehiscence	ProRoot MTA BP-RRM	1	65.8	94.3	88.2	Yes	Resorbable collagen membrane
von Arx et al., 2007 [14]	Prospective Cohort	194	191	43	Marginal bone level >3 mm	SuperEBA ProRoot MTA Retroplast	1	98.5	83.1	86.1	No	-
Kim E et al., 2008 [15]	Prospective Cohort	263	192	40	Kim and Kratchman	IRM SuperEBA ProRoot MTA	2	73.0	95.3	77.5	Yes	Calcium sulfate + resorbable collagen membrane (CollaTape)
Song et al., 2013 [51]	Prospective Cohort	199	135	33	Marginal bone loss >3 mm	SuperEBA ProRoot MTA	1–7	67.8	89.3	87.9	No	-
Song et al., 2013 [13]	Prospective Cohort	584	431	87	Kim and Kratchman	IRM SuperEBA ProRoot MTA	1–10	73.8	88.4	74.7	No	-
Song et al., 2011 [26]	Retrospective cohort	907	491	50	Kim and Kratchman	IRM SuperEBA ProRoot MTA	≥1	54.1	84.8	70.0	No	-
von Arx et al., 2012 [53]	Retrospective cohort	194	170	37	Crestal bone level >3mm	SuperEBA ProRoot MTA Retroplast	5	87.6	78.2	67.6	No	-
Song et al., 2012 [9]	Retrospective cohort	172	104	23	Kim and Kratchman	IRM SuperEBA ProRoot MTA	6–10	60.5	92.6	95.7	Yes	Resorbable collagen membrane (CollaTape)
Lui et al., 2014 [10]	Retrospective cohort	243	93	14	PD > 3 mm	IRM MTA	1–2	38	95.2	73.0	Yes	Resorbable collagen membrane (BioMend) + bone substitute (BioOss)
Song et al., 2018 [11]	Retrospective cohort	249	249	83	Kim and Kratchman	IRM SuperEBA ProRoot MTA	1	100	87.3	72.3	No	-
Kim D et al., 2020 [54]	Retrospective cohort	244	244	56	Kim and Kratchman	SCSM group: gray or white ProRoot MTA; FCSM group: RetroMTA or EndoCem MTA	1–6	100	94.7	71.4	No	-
Huang et al., 2020 [52]	Retrospective cohort	191	92 95 *	4 *	Preoperative PD >3 mm	IRM ProRoot MTA	5–9	48.2	80.8	50.0	Yes	Resorbable collagen membrane
Yoo et al., 2020 [17]	Retrospective cohort	652	225	9	Periodontal involvement	ProRoot MTA	5	34.5	82.4	33.3	Yes	BioOss

Endodontic-periodontal (EP); probing depth (PD); mineral trioxide aggregate (MTA); bioceramic paste root repair material (BP-RRM); super ethoxybenzoic acid (SuperEBA); dentine-bonded resin composite (Retroplast); zinc oxide-eugenol intermediate restorative cement (IRM); slow-setting calcium silicate–based materials (SCSM); fast-setting calcium silicate–based materials (FCSM); hydraulic calcium zirconia complex (RetroMTA); MTA-derived pozzolan cement (EndoCem MTA). * Specifically in this study the measure unit refers to root.

## Data Availability

Data will be made available upon request to the corresponding author.

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
