# Peer review of "Impact of Periodontal Attachment Loss on the Outcome of Endodontic Microsurgery: A Systematic Review and Meta-Analysis"

_medicina, 2021, doi:10.3390/medicina57090922_

Round 1

Reviewer 1 Report

This is a well written manuscript about an important topic relating to endodontic microsurgery success and the impact of periodontal attachment loss.

The authors could consider broadening the search terms as this may have contributed to missing RCT's or clinical studies. Furthermore the term outcomes could have been better defined in the methodology and discussion as to my knowledge there is no agreed core outcome set for endodontic success/failure and the reliance on ill defined clinical and radiographic success is imprecise. This could be added to the limitations.

The paper would have benefited by engaging a hospital/health librarian to validate the search methodology.

The discussion could include further explanation of the 2 discarded studies which showed greater success rates on teeth with attachment loss, relative to this paper's findings which showed the opposite. 

Overall this is a well written, well conceived and methodologically sound systematic review & meta-analysis worthy of publication, pending plagiarism software assessment.

Author Response

Reviewer #1:

  1. This is a well written manuscript about an important topic relating to endodontic microsurgery success and the impact of periodontal attachment loss.

Author’s response: We thank the reviewer #1 for the comments.

Revised text: Not applicable.

  1. The authors could consider broadening the search terms as this may have contributed to missing RCT's or clinical studies. Furthermore the term outcomes could have been better defined in the methodology and discussion as to my knowledge there is no agreed core outcome set for endodontic success/failure and the reliance on ill defined clinical and radiographic success is imprecise. This could be added to the limitations.

Author’s response: Once again, we thank the reviewer #1 comments. Regarding the search strategy, the final search key definition was preceded by a preliminary search in which the researchers assessed the broadest search terms (“endodontic microsurgery” AND outcome) in order to guarantee no relevant papers were excluded.

Additionally, as referred by the reviewer, criteria for assessing endodontic success/failure are highly variable among different studies. For this reason, the authors agreed to exclude all studies exhibiting unclear clinical and radiographic/clinical success criteria and narrow the inclusion to those which assessed success exclusively by following Rud and Molven’s outcome criteria, as mentioned in Table 1. Following the reviewer #1 valuable suggestion, and additional explanation of the outcome criteria of the included studies was presented at Materials and methods and also additionally dissected the outcome categories at Results section.

Revised text: “The included studies pursued Rud et al [28] and Molven et al [29] criteria for assessing of healing outcomes, with categories as follows: complete healing (re-establishment of the lamina dura); incomplete healing (the former radiolucency decreased in size or remained stationary with an irregular periphery); uncertain healing (the former radiolucency decreased in size or remained stationary with a circular periphery); and unsatisfactory healing (the former radiolucency increased in size). Outcome was dichotomized into success, when complete or incomplete healing was attributed, and failure, when healing was uncertain or unsatisfactory.“ – Manuscript, page 4.

“Six studies [8,9,15,17,52,53] report results with a breakdown of the four categories of outcome (complete, incomplete, uncertain and unsatisfactory healing), but in seven studies the success is reported without differentiation of complete and incomplete healing [10,11,13,14,51,55]. Incomplete healing reported in reviewed studies ranged from 5% [9,52,53] to 15% [17] and over 20% [8].” – Manuscript, page 6.

  1. The paper would have benefited by engaging a hospital/health librarian to validate the search methodology.

Author’s response: We thank the reviewer #1 for the risen point. However, given the cumulative experience our research group has on literature analysis and interpretation, a decision was made to have experienced researchers overviewing the search methodology.

Revised text: Not applicable.

  1. The discussion could include further explanation of the 2 discarded studies which showed greater success rates on teeth with attachment loss, relative to this paper's findings which showed the opposite. 

Author’s response: We thank the reviewer #1 comments which objective was to enhance the quality of the manuscript. In fact, only the study by Song et al. (2012) was examined in the Discussion section in regard to the greater success rate in teeth with attachment loss. Accordingly, analysis of the remaining paper (von Arx et al., 2007) was added.

Revised text: “Concerning the study of von Arx et al. [14], in which a higher success rate was reported for the periodontally involved teeth, it is worth mentioning that this is a preceding study to the one published in 2021 by the same group [53], which reported a lower success rate for the endodontic-periodontal group. We hypothesize that opposing conclusions derive from the difference in the follow-up period (1 vs 5 year). Also, we believe a longer follow-up time [53] to provide more relevant data than the one-year control [14]. This evidence highlights the importance of a sufficiently long follow-up period to detect the outcome of interest, and echoes that healing peaks in the first year after EMS, and a reversal to disease occurs in 5% to 25% of the apparently healed cases within 4 years after treatment [54].– Manuscript, page 11.

  1. Overall this is a well written, well conceived and methodologically sound systematic review & meta-analysis worthy of publication, pending plagiarism software assessment.

Author’s response: We thank reviewer #1 for the feedback.

Revised text: Not applicable.

Reviewer 2 Report

The paper is very well organized and statistical analysis is succesfully performed.

Author Response

(The authors gave the same response as above.)
